# Inherited Retinal Degeneration: Towards the Development of a Combination Therapy Targeting Histone Deacetylase, Poly (ADP-Ribose) Polymerase, and Calpain

**DOI:** 10.3390/biom13040581

**Published:** 2023-03-23

**Authors:** Yujie Dong, Jie Yan, Ming Yang, Wenrong Xu, Zhulin Hu, François Paquet-Durand, Kangwei Jiao

**Affiliations:** 1Key Laboratory of Yunnan Province, Yunnan Eye Institute, Affiliated Hospital of Yunnan University, Yunnan University, 176 Qingnian, Kunming 650021, China; 2Kunming Medical University, No. 1168 Chunrongxi Road, Chenggong District, Kunming 650500, China; 3Institute for Ophthalmic Research, University of Tübingen, 72076 Tübingen, Germany; 4Graduate Training Centre of Neuroscience, University of Tübingen, 72076 Tübingen, Germany; 5Department of Ophthalmology, First Affiliated Hospital of Kunming Medical University, Kunming 650032, China

**Keywords:** retinitis pigmentosa, cGMP, PKG, photoreceptor cell death

## Abstract

Inherited retinal degeneration (IRD) represents a diverse group of gene mutation-induced blinding diseases. In IRD, the loss of photoreceptors is often connected to excessive activation of histone-deacetylase (HDAC), poly-ADP-ribose-polymerase (PARP), and calpain-type proteases (calpain). Moreover, the inhibition of either HDACs, PARPs, or calpains has previously shown promise in preventing photoreceptor cell death, although the relationship between these enzyme groups remains unclear. To explore this further, organotypic retinal explant cultures derived from wild-type mice and *rd1* mice as a model for IRD were treated with different combinations of inhibitors specific for HDAC, PARP, and calpain. The outcomes were assessed using in situ activity assays for HDAC, PARP, and calpain, immunostaining for activated calpain-2, and the TUNEL assay for cell death detection. We confirmed that inhibition of either HDAC, PARP, or calpain reduced *rd1* mouse photoreceptor degeneration, with the HDAC inhibitor Vorinostat (SAHA) being most effective. Calpain activity was reduced by inhibition of both HDAC and PARP whereas PARP activity was only reduced by HDAC inhibition. Unexpectedly, combined treatment with either PARP and calpain inhibitors or HDAC and calpain inhibitors did not produce synergistic rescue of photoreceptors. Together, these results indicate that in *rd1* photoreceptors, HDAC, PARP, and calpain are part of the same degenerative pathway and are activated in a sequence that begins with HDAC and ends with calpain.

## 1. Introduction

Inherited retinal degeneration (IRD) represents a genetically diverse group of blinding diseases characterized by photoreceptor cell death [1]. The most common disease within the IRD group is retinitis pigmentosa (RP) [2], which affects about 1 in 4000 individuals worldwide [3]. Overall, IRD-type diseases remain untreatable, creating an important medical need [4]. Cyclic guanosine-3′, 5′-cyclic monophosphate (cGMP) has been found to play a central role in many genetically distinct types of IRD [5]. A mutation-induced upregulation of cGMP is directly or indirectly associated with the activities of protein kinase G (PKG), cyclic nucleotide-gated channels (CNGCs), histone-deacetylases (HDACs), poly-ADP-ribose-polymerases (PARPs), and calpain-type proteases [5,6,7,8]. The *rd1* mouse (retinal degeneration 1) is a naturally occurring IRD mouse model first described by Keeler in the early 1920s [9]. In these animals, the gene encoding for the beta subunit of the rod photoreceptor-specific phosphodiesterase-6 (PDE6) is mutated [10], causing PDE6 dysfunction, accumulation of cGMP in rod photoreceptors, and primary rod cell death, followed by secondary cone photoreceptor cell loss [11,12]. The degeneration of rods in the *rd1* mouse is associated with a prominent activation of HDAC, PARP, and calpain [13,14,15]. In humans, 5% to 6% of IRD patients carry mutations in PDE6 genes [8], making the *rd1* mouse a relevant animal model for studies into possible treatments. 

HDACs form a group of enzymes that catalyze the removal of acetyl groups from lysine residues of both histone and nonhistone proteins. In humans, there are 18 different HDAC enzymes that use either zinc- or NAD^+^-dependent mechanisms to deacetylate acetyl-lysine substrates and that are grouped into four main classes. HDAC activity has been found to be involved in cGMP–PKG signaling-induced photoreceptor degeneration [13,16], and inhibition of HDAC prevented cGMP-dependent neurodegeneration in several IRD animal models [17,18,19]. While the mechanism of HDAC activation remains unclear, HDACs have been suggested to act upstream of PARP activity during photoreceptor degeneration [13]. PARPs belong to a diverse class of enzymes that catalyze ADP-ribose transfer to target proteins and are associated with DNA damage repair [20,21]. PARP can sequentially add ADP-ribose units from NAD^+^ to form polymeric ADP-ribose (PAR) chains [22]. PARP may also be the primary driver for a specific form of cell death, termed PARthanatos [23]. In line with this, excessive PARP activity was found to contribute to photoreceptor degeneration [24,25,26]. Furthermore, PARPs may also be involved in regulating calpain activity, possibly via changes in photoreceptor energy metabolism and Ca^2+^ homeostasis [25]. Calpains relate to a class of Ca^2+^-activated non-lysosomal neutral proteases involved in a broad range of cellular functions [27]. In the *rd1* mouse, a cGMP-dependent overactivation of CNG channels may cause excessive Ca^2+^ influx and subsequent overactivation of calpains [28,29]. Accordingly, excessive calpain activity has been implicated in retinal degeneration, and inhibition of calpain was found to be beneficial in different IRD animal models [30,31].

Taken together, HDAC, PARP, and calpain have all been suggested to be therapeutic targets for the treatment of IRD [13,24,25]. Notably, HDAC and PARP activities, on the one hand, and CNGC and calpain activity, on the other hand, have been proposed to form part of two branches of the same cGMP-dependent cell death pathway [8]. If this view is correct, then the selective inhibition of both pathway branches should synergistically protect photoreceptors from mutation-induced degeneration. To explore this hypothesis, we used the HDAC inhibitor suberoylanilide hydroxamic acid (SAHA; INN: Vorinostat) [32,33], the PARP inhibitor Olaparib (OLA) [34], and the calpain inhibitor Calpastatin [35], as well as combinations of these. Contrary to our expectations, we found that (1) HDAC controlled both PARP and calpain activities, and that (2) HDAC, PARP, and calpain likely were constituent members of the same degenerative pathway. These results may guide the design and development of future therapeutic approaches for the treatment of IRD.

## 2. Materials and Methods

### 2.1. Animals

In the present study, we used C3H *Pde6b^rd1/rd1^* mice (*rd1*), and congenic C3H *Pde6b*^+/+^ animals as wild-type (*wt*) controls. Animals were housed under standard white cyclic lighting, had free access to food and water, and were used irrespective of gender. The study was conducted according to the ARVO statement for the use of animals in ophthalmic and vision research and complied with the regulations of German law on animal protection. Experiments performed in China were reviewed and approved by the Yunnan University ethical review board (No. YNUCARE20210024), while experiments performed in Germany were approved by the Tübingen University committee on animal protection (Einrichtung für Tierschutz, Tierärztlicher Dienst und Labortierkunde) and registered under No. AK02/19M. All efforts were made to minimize the number of animals used and their suffering.

### 2.2. Retinal Explant Culture Procedure

Retinal culture was performed as previously published [36]. In brief, animals were sacrificed by decapitation at post-natal day 5. After cleaning the heads with 70% ethanol, the eyes were removed under aseptic conditions and placed into R16 basal medium (No. 074-90743A, Gibco, Paisley, UK). After a 5 min wash, the eyes were pre-digested for 8–10 min in 0.12% Proteinase K (193504; MP Biomedicals, Illkirch-Grafenstaden, France) at 37 °C to allow for separation of the retinal pigment epithelium from the sclera. Proteinase K was then inactivated by placing the eyes in R16 basal medium with 10% fetal bovine serum (FSD501; Gemini, West Sacramento, CA, USA) for 2 min. Under sterile conditions and using a stereomicroscope (Carl Zeiss Microscopy GmbH, Oberkochen, Germany), the anterior segment, lens, and vitreous body were carefully removed from the eyeballs. The optic nerve was then cut. Afterwards, the retinas were separated from the sclera, together with the attached retinal pigment epithelium. Then, the retinas were cut into four wedge shapes. Next, the retinas were transferred to cell culture plate inserts (CLS3412; Corning-Costar, Grand Rapids, MI, USA), with the ganglion cell layer facing upwards. Subsequently, the membrane was placed in a six-well culture plate and incubated in 1.5 mL of R16 medium plus supplements [36] (i.e., complete medium) at 37 °C in a humidified incubator under 5% CO_2_.

### 2.3. Retinal Drug Treatments

The explanted retinas were cultured in R16 medium with supplements for 2 days without treatment to adapt to the in vitro conditions. From P7 onwards, they were treated with different inhibitors: 0.1 μM SAHA (HY-10221; MedChemExpress, Sollentuna, Sweden), 1 μM Olaparib (HY-10162; MedChemExpress, Sollentuna, Sweden), and 20 μM Calpastatin (SCP0063; Sigma-Aldrich, St. Louis, MO, USA) for monotherapies and 0.1 μM SAHA and 20 μM CAST or 1 μM Olaparib and 20 μM CAST for combined therapies. There were at least five retinal explants from different animals in each group. During the culturing period, the R16 medium was changed every 2 days and the cultures were terminated at PN11 by fixation with 4% paraformaldehyde (PFA) or by direct freezing in liquid nitrogen (unfixed preparations). Afterwards, the retinas were embedded in O.C.T^TM^ compound (4583; SAKURA, Alphen aan den Rijn, the Netherlands) and stored at −20 °C. The retina blocks were sectioned sagittally with 12 µm thickness using a microtome (Thermo Fisher Scientific, CryoStar NX50 OVP, Runcorn, UK) and collected on Superfrost Plus glass slides (R. Langenbrinck, Emmendingen, Germany). The glass slides were stored at −20 °C for further experiments.

### 2.4. Detection of Cell Death

A terminal dUTP nick-end labelling (TUNEL) assay kit (Roche Diagnostics, Mannheim, Germany) was used to detect dying cells in retinal tissue sections. Histological sections from retinal explants were dried and stored at −20 °C. The sections were rehydrated with phosphate-buffered saline (PBS; 0.1 M) and incubated with proteinase K (1.5 µg/µL) diluted in 50 mM TRIS-buffered saline (TBS; 1 µL enzyme in 7 mL TBS) for 5 min. This was followed by washing in TBS 3 times for 5 min each and incubation with blocking solution (10% normal goat serum, 1% bovine serum albumin, and 1% fish gelatin in phosphate-buffered saline with 0.03% Tween-20). TUNEL staining solution was prepared using 10 parts of blocking solution, 9 parts of TUNEL labelling solution, and 1 part of TUNEL enzyme. After blocking, the sections were incubated with TUNEL staining solution overnight at 4 °C. Finally, the sections were washed twice with PBS, mounted using Vectashield with DAPI (ab104139; Abcam, Cambridge, UK), and imaged under a Zeiss ApoTome 2 microscope for further analysis.

### 2.5. HDAC In Situ Activity Assay

The HDAC activity assay was performed on cryosections of 4% PFA-fixed eyes in both *rd1* and *wt* animals. The assay is based on an adaptation of the FLUOR DE LYS^®^-Green System (Biomol, Hamburg, Germany). Retina sections were exposed to 50 µM FLUOR DE LYS^®^-SIRT1 deacetylase substrate (BML-Kl177-0005; ENZO, New York, NY, USA) with 2 mM NAD^+^ (BML-KI282-0500; ENZO, New York, NY, USA) in assay buffer (50 mM Tris/HCl, 137 mM NaCl; 2.7 mM KCl; 1 mM MgCl_2_; pH 8.0) for 3 h at room temperature. Sections were then washed in PBS and fixed in methanol at −20 °C for 20 min. Slides were mounted with FLUOR DE LYS^®^ developer II concentrate (BML-KI176-1250; Enzo, New York, NY, USA), which was diluted 1:5 in assay buffer. Note that the HDAC activity assay detects activities of class I, II, and III HDACs [13].

### 2.6. Calpain In Situ Activity Assay 

Unfixed retinal tissue sections were incubated and rehydrated for 15 min in calpain reaction buffer (CRB) (5.96 g HEPES, 4.85 g KCl, 0.47 g MgCl_2_, and 0.22 g CaCl_2_ in 100 mL ddH_2_O; pH 7.2) with 2 mM dithiothreitol (48315; Sigma-Aldrich, St. Louis, MO, USA). The tissue sections were incubated for 2.5 h at 37 °C in CRB with tBOC-Leu-Met-CMAC (25 μM; A6520; Thermo Fisher Scientific, Runcorn, UK), then washed with PBS and incubated with ToPro (1:1000 in PBS, T3605; Thermo Fisher Scientific, Runcorn, UK) for 20 min. Afterwards, the tissue sections were washed in PBS and mounted using Vectashield without DAPI (H-1000; Vector, Burlingame, CA, USA) for immediate visualization using a Zeiss Axio Imager M2.

### 2.7. Calpain-2 Immunostaining

Tissue sections from fixed retina were dried at 37 °C for 1 h, then the slides were washed with PBS for 15 min. This was followed by application of blocking solution (10% NGS, 1% BSA, and 0.3% PBST) for 1 h at room temperature. Then, the calpain-2 primary antibody diluted in blocking solution was applied for overnight incubation at 4 °C (1:200, ab39165; Abcam, Cambridge, UK). The sections were washed 3 × 10 min with PBS. Goat anti-rabbit secondary antibody diluted in PBS (1:300, ab175471; Abcam, Cambridge, UK) was then applied for 1 h at room temperature. The sections were further washed with PBS 3 × 10 min and mounted with mounting medium with DAPI (ab104139; Abcam, Cambridge, UK).

### 2.8. PARP Activity Assay

Unfixed retinal tissue sections were incubated and rehydrated for 10 min in PBS. The reaction mixture (10 mM MgCl_2_, 1 mM dithiothreitol, and 50 µM 6-Fluo-10-NAD^+^ (N 023; Biolog, CA, USA) in 100 mM Tris buffer with 0.2% Triton X100, pH 8.0) was applied to the sections for 3 h at 37 °C. After three washed in PBS for 5 min each, the sections were mounted in Vectashield with DAPI (H-1200; Vector) for immediate visualization with a Carl Zeiss Axio Imager M2.

### 2.9. Western Blot

After a culture period of six days, the retinas were collected, homogenized, and protein was isolated using cell extraction buffer (FNN0011; Thermo Fischer Scientific, Karlsruhe, Germany) containing Protease Inhibitor Cocktail Set III (539134; Merck, Darmstadt, Germany) and 1 mM phenylmethylsulfonylfluoride (6367.4; Carl Roth, Karlsruhe, Germany). The protein concentration was determined using the BCA Protein Assay Kit (23227; Thermo Fischer Scientific, Karlsruhe, Germany). Equal amounts of protein (10 µg) were loaded onto 12% SDS gels, followed by wet-transfer with Towbin buffer to nitrocellulose membranes. SDS-PAGE and wet-transfer were performed using standard protocols [37]. Immunostaining was carried out with a PARP-1 antibody (1:2000, ab32138; Abcam, Cambridge, UK), with α-Tubulin antibody (1:1000; ab24610; Abcam, Cambridge, UK) used for normalization. Secondary antibodies were incubated simultaneously for 1 h at RT in the dark. The images were visualized using the Odyssey Fc Imaging System (LI-COR Biosciences, Lincoln, NE, USA).

### 2.10. Image and Data Analysis

For western blot images, densitometry was performed by acquiring signals from single protein bands, using Image Studio Lite Software (LI-COR) and delimiting each protein band. To calculate a normalized PARP-1 protein concentration for each band, the signals from the housekeeping proteins were normalized, where the observed signal for the housekeeping protein of each lane was divided by the highest observed signal of the housekeeping protein on the blot. Therefore, each lane had a lane normalization factor (LNF), allowing for comparison between biological replicates. The signal obtained from each PARP-1 band was divided by the corresponding LNF and this was used as the final normalized experimental signal (NES). For presentation of the results, the untreated control condition (*rd1*) was normalized to one, with all other groups presented as a percentage of the control.

The images of organotypic explant cultures were captured using a Zeiss Imager Z.2 fluorescence microscope, equipped with ApoTome 2, an Axiocam 506 mono camera, and an HXP-120V fluorescent lamp (Carl Zeiss Microscopy, Oberkochen, Germany). The excitation (λExc)/emission (λEm) characteristics of the filter sets used for the different fluorophores were as follows (in nm): DAPI (λExc = 369 nm, λEm = 465 nm), AF488 (λExc = 490 nm, λEm = 525 nm), AF568 (λExc = 578 nm, λEm = 602 nm), and ToPro (λExc = 642 nm, λEm = 661 nm). Zen 2.3 blue edition software (Zeiss) was used to capture the images (tiled and z-stack, 20× magnification). Sections of 12 µm thickness were analyzed using 12 Apotome Z-planes. Data were obtained from retinal explants derived from at least five different mice for each experimental group. For the quantification of positive cells in the retinal outer nuclear layer (ONL), we proceeded as follows: The number of cells in six different rectangular ONL areas was counted manually based on the number of DAPI-stained nuclei and used to calculate an average ONL cell size. This average ONL cell size was then used to calculate the total number of cells in a given ONL area. The percentage of positive cells was calculated by dividing the absolute number of positive cells by the total number of ONL cells. Adobe Photoshop 2022 and Adobe Illustrator 2022 (Adobe Systems Incorporated, San Jose, CA, USA) were used for image processing. An example of this percent quantification is given for the TUNEL assay in Appendix A Figure A2. To be able to compare the various readouts (i.e., TUNEL, HDAC, PARP, calpain, calpain-2), data points were normalized such that the lowest percent values were set to zero (i.e., in *wt* retina) whereas the highest percent values were set to one (i.e., in *rd1* retina). The formula used for normalization by linear scaling was: χscaled = χ – χmin / χmax − χmin, using SPSS Statistics 26 (IBM, Armonk, New York, NY, USA). Note that normalization was not used for western blot data. For statistical analysis of the data, an ordinary one-way ANOVA with Tukey multiple comparison test was performed to account for repeated measurements. Unpaired t-test was used for comparison between two groups and all bars are shown with standard deviation. All calculations were performed with GraphPad Prism 8 (GraphPad Software, La Jolla, CA, USA); *p* < 0.05 was considered significant. Levels of significance indicated in figures are as follows: *, *p* < 0.05; **, *p* < 0.01; ***, *p* < 0.001; ****, *p* < 0.0001.

## 3. Results

We used retinal explant cultures derived from *rd1* mice and cultured from post-natal day (P) 5 to 11 to evaluate the effects of either single drug treatment (i.e., monotherapy) or combined drug treatment (i.e., combination therapy) on photoreceptor cell death. For comparison, we also included retinal explant cultures derived from wild-type (*wt*) mice. To target HDAC, PARP, and calpain, respectively, we used the highly specific inhibitors suberoylanilide hydroxamic acid (SAHA), N-acylpiperazine Olaparib (OLA), and Calpastatin (CAST), and combinations of these. As readouts, we used the TUNEL assay for the detection of dying cells and in situ activity assays to resolve the cellular activities of HDAC, PARP, and calpain. Since the activity of the calpain-2 isoform has previously been related to neurodegenerative diseases [38], we used immunolabeling specifically for activated calpain-2 as an additional readout.

### 3.1. Monotherapy and Combination Therapy Both Delay rd1 Photoreceptor Degeneration

Initially, we tested the effects of SAHA on *rd1* mouse retinal degeneration, using two different compound concentrations. To account for varying experimental conditions, data points were normalized such that the lowest cell death rate was set to zero (i.e., in *wt* retina) whereas the highest cell death value was set to one (i.e., in *rd1* retina). In the untreated *rd1* retina, at P11 in vitro, the normalized value for cells dying in the outer nuclear layer (ONL) was elevated at 0.77 (±0.13, *n* = 27). In comparison, treatment with 0.1 µM SAHA reduced cell death in the *rd1* ONL to 0.32 (±0.1, *n* = 18, *p* < 0.0001). However, at 1 µM SAHA, photoreceptor degeneration was increased to 0.53 (±0.11, *n* = 5, *p* = 0.0004) (Figure A1). At the same time, the thickness of the ONL as a measure of photoreceptor survival was essentially halved from 37.54 µm (±2.82, *n* = 5) at 0.1 µM SAHA to 18.12 µm (±4.23, *n* = 5, *p* < 0.0001) at 1 µM SAHA. Taken together, this data suggested 0.1 µM SAHA as an appropriate dose for further studies on *rd1* treatment while also indicating that SAHA´s pharmaceutical window in the *rd1* retina was relatively small. 

TUNEL-positive cells in the ONL were significantly increased in *rd1* mice (0.77 ± 0.13, *n* = 27) compared to the *wt* situation (0.1 ± 0.06, *n* = 12, *p* < 0.0001) (Figure 1). Monotherapy with CAST (0.43 ± 0.18, *n* = 14, *p* < 0.0001), SAHA (0.32 ± 0.1, *n* = 18, *p* < 0.0001), or OLA (0.51 ± 0.17, *n* = 17, *p* < 0.0001) significantly reduced TUNEL positivity in the *rd1* ONL. Similarly, combination therapy with SAHA and CAST (0.4 ± 0.11, *n* = 6, *p* < 0.0001) or OLA and CAST (0.49 ± 0.13, *n* = 15, *p* < 0.0001) reduced ONL cell death. Interestingly, the SAHA monotherapy group showed the strongest protective effect on *rd1* photoreceptor degeneration whereas the two combination therapies showed no additional rescue effect compared to the respective monotherapies. To investigate whether a combination therapy might have adverse effects in the *wt* situation, we tested SAHA and CAST together on the P11 *wt* retina, albeit without finding evidence for toxicity in either the ONL or the inner nuclear layer (INL) (Appendix A Figure A2).

### 3.2. Monotherapy and Combination Therapy Reduce Calpain Activity

To investigate how HDAC and PARP activities might interact with calpain, we used specific inhibitors and investigated calpain activity using a general in situ activity assay [39] as well as immunolabeling for activated calpain-2 [40]. While calpain activity was rather low in the *wt* retina (calpain activity: 0.06 ± 0.05, *n* = 12; calpain-2 activation: 0.14 ± 0.13, *n* = 10), it was strongly and significantly increased in the *rd1* ONL (calpain activity: 0.69 ± 0.21, *n* = 22, *p* < 0.0001; calpain-2 activation: 0.67 ± 0.16, *n* = 21, *p* < 0.0001) (Figure 2). The specific calpain inhibitor CAST significantly decreased both overall calpain activity and calpain-2 activation in *rd1* photoreceptors (calpain activity: 0.14 ± 0.11, *n* = 13, *p* < 0.0001; calpain-2: 0.38 ± 0.12, *n* = 14, *p* < 0.0001). General calpain activity and calpain-2 activation were also significantly reduced after monotherapy with the HDAC inhibitor SAHA (calpain activity: 0.17 ± 0.07, *n* = 15, *p* < 0.0001; calpain-2: 0.35 ± 0.2, *n* = 10, *p* < 0.0001), the PARP inhibitor OLA (calpain activity: 0.29 ± 0.06, *n* = 9, *p* < 0.0001; calpain-2: 0.37 ± 0.17, *n* = 10, *p* < 0.001), as well as by combined treatment with SAHA and CAST (calpain activity: 0.31 ± 0.07, *n* = 5, *p* < 0.0001; calpain-2: 0.4 ± 0.1, *n* = 6, *p* < 0.05) or OLA and CAST (calpain activity: 0.26 ± 0.09, *n* = 8, *p* < 0.0001; calpain-2: 0.32 ± 0.16, *n* = 6, *p* < 0.001) (Figure 2). Taken together, these results suggested that calpain activation was secondary to both HDAC and PARP activities.

### 3.3. HDAC Activity Is Not Affected by PARP or Calpain Inhibition

To investigate the potential effects of PARP and calpain activities on HDAC, an HDAC in situ activity assay was performed after monotherapy and combination therapy. Compared to the *wt* ONL (0.05 ± 0.03, *n* = 12), HDAC activity in the *rd1* ONL was significantly increased (0.53 ± 0.21, *n* = 20, *p* < 0.0001) (Figure 3). Treatment with the calpain inhibitor CAST had no significant effect on HDAC-positive cells in the *rd1* ONL (0.46 ± 0.16, *n* = 16, *p* > 0.05). As a broad-spectrum inhibitor of class I and II HDACs, SAHA strongly reduced *rd1* HDAC activity (0.25 ± 0.09, *n* = 11, *p* < 0.001) whereas treatment with the PARP inhibitor OLA had no effect on the number of HDAC-positive cells in the *rd1* ONL (0.63 ± 0.18, *n* = 10, *p* > 0.05). Combined treatment with SAHA and CAST reduced *rd1* HDAC activity to a level comparable to SAHA monotherapy (0.15 ± 0.03, *n* = 6, *p* < 0.0001). Remarkably, combination therapy with OLA and CAST significantly reduced *rd1* HDAC activity beyond the individual drug treatments (0.29 ± 0.15, *n* = 7, *p* < 0.05).

### 3.4. PARP Activation Depends on HDAC but Not on Calpain Activity

To further evaluate the interactions between HDAC, PARP, and calpain, a PARP in situ activity assay was performed on retinal explant cultures. Compared to the *wt* ONL (0.11 ± 0.07, *n* = 9, *p* > 0.05), PARP activity was significantly increased in the *rd1* situation (0.64 ± 0.19, *n* = 20, *p* < 0.0001) (Figure 4). Compared to the *rd1* untreated control, the calpain inhibitor CAST did not reduce the number of ONL-positive cells for PARP activity (0.65 ± 0.15, *n* = 10, *p* > 0.05). Since earlier research indicated that PARP, notably the PARP-1 isoform, could be a target for calpain-dependent proteolytic cleavage [41], we investigated possible PARP-1 fragmentation using western blotting. However, we found no evidence for PARP-1 cleavage either in untreated or in CAST-treated retinal samples (Figure A3).

Monotherapy with the HDAC inhibitor SAHA significantly reduced *rd1* PARP activity (0.27 ± 0.12, *n* = 15, *p* < 0.0001) (Figure 4). As expected, the strongest decrease in *rd1* PARP activity was observed with the PARP-specific inhibitor OLA (0.1 ± 0.04, *n* = 9, *p* < 0.0001). Both combined treatment with either SAHA and CAST (0.32 ± 0.16, *n* = 6, *p* < 0.001) or OLA and CAST (0.18 ± 0.13, *n* = 8, *p* < 0.0001) decreased the activity of PARP in the *rd1* ONL to levels comparable to the respective monotherapies. 

## 4. Discussion

The enzymatic activities of HDAC, PARP, and calpain have been suggested to contribute to the pathogenesis of IRD [5] and feature prominently in cGMP-induced photoreceptor degeneration. However, it is still unclear whether these three groups of enzymes act in a specific sequence or whether they promote cell death independently from each other. Our results indicated that HDAC, PARP, and calpain act in concert within the same cell death pathway. We also found that the non-selective HDAC inhibitor SAHA showed a remarkable therapeutic effect, albeit only in a relatively narrow therapeutic range.

### 4.1. Histone Deacetylases and Photoreceptor Degeneration

Experimental evidence indicates that excessive HDAC activation leads to photoreceptor degeneration in various IRD models, but the exact roles of individual HDAC isoforms during the degenerative process are still largely unclear. For instance, the activities of class I and II HDACs were related to inherited photoreceptor cell death whereas class III HDACs (aka sirtuins) did not seem to be involved [13]. Overexpression of HDAC4 in the *rd1* mouse was found to prolong photoreceptor survival, and reduced expression of HDAC4 in wild-type retinas was correlated with increased photoreceptor cell death [42]. Furthermore, the broad-spectrum HDAC inhibitors trichostatin A (TSA), valproic acid (VPA), and scriptaid can protect *rd1* photoreceptors from cell death [43,44]. However, VPA accelerated photoreceptor degeneration in the *rd10* mouse model for IRD [45], indicating the complexity of HDAC functions during retinal degeneration. 

SAHA is a non-selective broad-spectrum inhibitor of HDACs. It was approved for the treatment of cutaneous T-cell lymphoma in 2006 [46] and has entered clinical trials for the treatment of other diseases, such as breast cancer [47]. Our study revealed that SAHA reduced photoreceptor cell death, in line with a previous study on the *cpfl1* mouse cone photoreceptor degeneration model [43]. In addition, monotherapy with SAHA was more effective in preventing photoreceptor cell death than treatment with OLA and CAST. Moreover, combined treatment with either SAHA and CAST or OLA and CAST did not surpass SAHA´s protective effect. However, the therapeutic window of SAHA appeared to be small, with obvious signs of toxicity already observed at 1 µM concentration. Given the difficulties in achieving retinal drug delivery and appropriate long-term dosing [48], this may limit the applicability of SAHA for the treatment IRD. 

### 4.2. PARP as a Therapeutic Target in cGMP-Dependent Cell Death

PARP-type enzymes belong to an 18-member family involved in many different functions, including DNA damage repair and regulation of gene transcription [49]. Within the PARP family, PARP-1 is the best-characterized member [50] and its excessive activation has been connected with neurodegenerative diseases [14,51] and a particular form of cell death, termed PARthanatos [23]. In several animal models of photoreceptor cell death, including *rd1* mice as well as P23H and S334ter rats, PARP was shown to be overactivated [30] while PARP inhibition or gene knockout resulted in neuroprotection [25,52]. Our previous experiments also revealed that excessive activation of PARP may occur prior to retinal degenerative disease and accelerate cell death [14,25,53]. This is in line with our current results showing that OLA attenuated photoreceptor degeneration. How exactly PARP activity may precipitate cell death is not fully understood, although the excessive consumption of its substrate NAD^+^ could cause mitochondrial dysfunction eventually leading to a cessation of ATP production and energy depletion [54]. Altogether, these findings further highlight PARP as a therapeutic target in cGMP-dependent photoreceptor cell death. 

### 4.3. Calpains as a Therapeutic Target in IRD

Calpains form a group of Ca^2+^-activated non-lysosomal neutral proteases, which includes 15 different isoforms [55]. The activation of calpains has been connected to neuronal cell death in a variety of different diseases [56] and likewise was found to promote photoreceptor cell death in different IRD models [30]. Our results in the *rd1* retina confirmed this, indicating that excessive calpain activity, and specifically calpain-2, takes part in cGMP-dependent cell death in IRD. Calpains possess numerous substrates, including cytoskeletal proteins, kinases, and phosphatases, membrane receptors and transporters, and steroid receptors [57], explaining how excessive calpain activation may promote cell death. 

Nevertheless, the question remains whether calpain activity can be viewed as a potential therapeutic target in retinal neurodegeneration. CAST is a highly specific endogenous calpain inhibitor with neuroprotective properties; however, it blocks several different calpain isoforms at the same time [58,59]. This is relevant as especially calpain-1 has been suggested to be neuroprotective whereas calpain-2 activity is believed to be detrimental [38,58]. Both calpain isoforms are inhibited by CAST. In our study, treatment could not block *rd1* retinal degeneration completely, suggesting that beneficial effects resulting from calpain-2 inhibition might have been offset by simultaneous calpain-1 inhibition. Hence, further research on the precise roles of different calpain isoforms during neurodegeneration is required, and future therapy approaches may benefit from the development of calpain-2-specific inhibitors. 

### 4.4. Molecular Interactions between HDAC, PARP, and Calpain

The enzymatic activities of HDAC, PARP, and calpain have been connected to IRD and specifically to cGMP-induced cell death [5]. However, the molecular interactions among HDAC, PARP, and calpain are still not completely understood. Previously, we found the HDAC inhibitor trichostatin A (TSA) significantly decreased PARP activity during *rd1* photoreceptor cell death [13]. This suggested that HDAC was regulating PARP activity in cGMP-induced cell death, an interpretation that was consistent with our current results obtained with SAHA. SAHA´s neuroprotective effect may stem from a downregulation of oxidative stress pathways [60]. During oxidative stress, excessive production of reactive oxygen species will cause DNA damage [61,62], leading to activation of DNA repair enzymes, including PARP [63,64]. Subsequently, PARP may indirectly activate calpains via its excessive consumption of NAD^+^ and a resultant breakdown of cellular energy metabolism [25]. Loss of intracellular ATP will impair the function of plasma membrane Ca^2+^ ATPase (PMCA), an active transporter that is crucial for keeping intracellular Ca^2+^ levels low [65]. Remarkably, HDAC inhibition has been found to increase PMCA activity [66,67], which thus could enhance intracellular Ca^2+^ clearance and reduce calpain activity. 

Previous research indicated that the cGMP-induced cell death pathway in *rd1* photoreceptors might have two branches, one governed by calpain activity and the other characterized by HDAC and PARP activities [5]. Therefore, we hypothesized that combined treatments targeting either HDAC and calpain or PARP and calpain together should synergistically improve photoreceptor cell viability. However, the combination therapies tested here did not produce the expected synergistic effects compared to the corresponding monotherapies. In principle, this lack of synergistic effect could be due to off-target effects, even though the inhibitors employed here have been extensively tested and previously validated (see Section 4.5 below). Additionally, it is worth keeping in mind that synergistic interaction may not necessarily occur between two drugs if one drug affects the therapeutic efficacy of the other [68]. However, given the very different targets and modes of actions of the inhibitors used here (i.e., CAST vs. SAHA or OLA), such drug interaction does not seem likely. Thus, overall, our data suggests that HDAC, PARP, and calpain are integral to the very same cell death pathway, offering a new framework for the understanding of photoreceptor cell death (Figure 5).

In a previous study, we found 5-methylcytosine (5mC) to accumulate in *rd1* photoreceptors and that inhibition of DNA methyltransferases (DNMTs) with the drug decitabine reduced photoreceptor cell death [26]. These findings highlighted excessive DNA methylation as a contributing factor for retinal degeneration. HDAC inhibition is capable of eliciting changes in DNA methylation [69], and PARP and calpain are also reported to affect DNA methylation by interfering with downstream ten-eleven translocation (TET) protein-mediated DNA hydroxy-methylation [70,71]. Hence, for future studies into possible combination therapies, it may be interesting to combine DNMT inhibitors such as decitabine on the one hand with inhibition of calpain or HDAC/PARP on the other hand. Moreover, very recently we found the transcription factor early-growth-response-1 (EGR-1) to be an important upstream mediator of photoreceptor degeneration [72]. While there are currently no pharmacological inhibitors available to regulate EGR-1 activity, downregulation of EGR-1 using molecular approaches could be combined with inhibition of either HDAC, PARP, or calpain.

### 4.5. Towards the Identification of the HDAC, PARP, and Calpain Isoforms Responsible for Photoreceptor Degeneration

It has been more than a decade that the activities of HDAC, PARP, and calpain have been connected to the progression of photoreceptor loss in IRD [13,14,15]. Since each of these three groups of enzymes contains a multitude of isoforms (18 different genes/isoforms for HDAC, 17 for PARP, 15 for calpains), it is to date not clear which of these isoforms is precisely responsible for photoreceptor degeneration. While a molecular biology approach could potentially resolve this question, given the number of different genes that would need to be manipulated (i.e., 50 different genes in total) and the difficulties in selectively manipulating a given gene without introducing compensatory regulation [73], such molecular studies are an undertaking that is currently beyond the capabilities of most academic laboratories.

Therefore, in the present study, we resorted to pharmacology and the use of three particularly well-characterized and validated inhibitors, SAHA, OLA, and CAST. Notably, SAHA inhibits HDAC-1, -2, and -3 in the low nanomolar range while other HDAC isoforms are inhibited only in the micromolar range [33,74,75]. Hence, our study using 100 nM SAHA pointed towards HDAC-1, -2, and -3 as the HDAC isoforms most likely to be responsible for photoreceptor degeneration. This was in line with other studies indicating that HDAC-4 [42] and HDAC-6 [76] may promote survival of retinal neurons including photoreceptors. The possible protective effects of HDAC-4 and HDAC-6 may also explain why higher concentrations of SAHA were detrimental in our study.

The drug OLA blocks PARP-1, PARP-2, and tankyrase1 with IC_50_ values of 5 nM, 1 nM, and 1.5 μM, respectively [34]. This makes it seem likely that in our study with 1 µM OLA, PARP-1 and PARP-2 were completely inhibited while other PARP-like proteins may have been less affected. PARP-1 appears to be responsible for ≈ 90% of cellular PARP activity [77] and a previous study from our group found PARP-1 knockout retina to be resistant against cGMP-induced retinal degeneration [52]. Together, this suggests PARP-1 as a likely candidate for promoting photoreceptor cell death, even though a contribution from PARP-2 cannot be excluded at this point.

The calpain inhibitor CAST is derived from the peptide sequence of calpastatin, the endogenous inhibitor of calpains [78]. As such, CAST is known to block the activities of calpain-1, -2, -3, -8, and -9 [79] and its inhibitory effect on these isoforms is in the nanomolar range [80]. Thus, a concentration of 20 μM likely produced a complete block of calpain-1, -2, -3, -8, and -9. However, calpain-8 and -9 are preferentially expressed in the gastrointestinal tract [79]. Calpain-1 and -2 are ubiquitously expressed in most tissues, including in the retina, which may also display calpain-3 expression [15]. In neurons, calpain-1 and -2 were found to play opposite roles such that calpain-1 was neuroprotective while calpain-2 was degenerative [59]. Since in our present study we found calpain-2 to be over-activated compared to the wild-type, calpain-2 may be the isoform most likely to be involved in photoreceptor cell death.

Taken together, the analysis of our pharmacological data may allow us to narrow down the number of HDAC, PARP, and calpain isoforms responsible for photoreceptor degeneration to a relatively small group, which may be manageable for future molecular studies. This group of isoforms of interest comprises at least HDAC-1, HDAC-2, HDAC-3, PARP-1, PARP-2, and calpain-2. This relatively small group of isoforms may also be targeted in pharmacological studies with more specific inhibitors, as soon as such may become available.

## 5. Conclusions

Our results shed new light on the degenerative pathways triggered by high cGMP in *rd1* photoreceptors by proposing a sequential activation of HDAC, PARP, and calpain. The position of HDAC at the beginning of this degenerative sequence may explain the relatively strong neuroprotective efficacy of HDAC inhibition. While the combination therapies tested here did not produce the desired synergistic effect, our observations may impact the design of future studies. For instance, addressing other drug target combinations, such as DNAMT and calpain-2 or EGR-1 and PARP, could potentially reveal synergistic photoreceptor neuroprotection. Taken together, our findings may provide a basis for the development of an effective combination therapy for IRD.

## Figures and Tables

**Figure 1 biomolecules-13-00581-f001:**
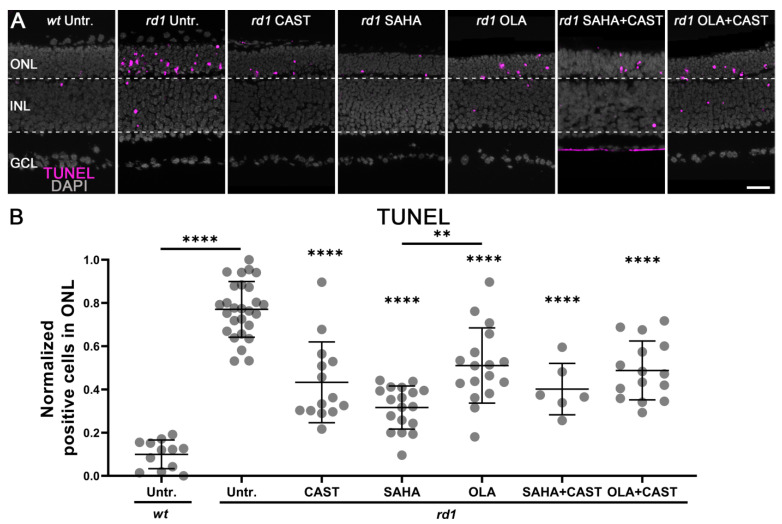
Effects of mono- and combination therapies on photoreceptor cell death. Retinal explant cultures derived from either wild-type (*wt*) or *rd1* animals were treated until post-natal day 11 with specific inhibitors targeting calpain (CAST), HDAC (SAHA), or PARP (OLA), or were treated with combinations of these inhibitors. (**A**) Retinal cross-sections obtained from treated or untreated (Untr.) specimens were stained with the TUNEL assay (magenta) to detect dying cells. DAPI (grey) was used as nuclear counterstain. (**B**) Normalized quantification (*rd1* max value = 1, *wt* min value = 0) of cell death in the outer nuclear layer (ONL) in the various treatment groups. Note the strong increase in ONL cell death in *rd1* retina compared to *wt* and the decreased cell death rates afforded by the various treatments. Combined treatment with either SAHA + CAST or OLA + CAST did not produce additional rescue effects compared to single drug treatment. Images shown represent at least six different specimens for each genotype. Untr. *wt*: 12 explants from 12 different mice; untr. *rd1*: 27/27; CAST *rd1*: 14/14; SAHA *rd1*: 18/18; OLA *rd1*: 17/17; SAHA + CAST *rd1*: 6/6; OLA + CAST *rd1*: 15/15; error bars represent SD; INL = inner nuclear layer; GCL = ganglion cell layer. Scale bar = 50 µm; significance levels: **, *p* < 0.01; ****, *p* < 0.0001.

**Figure 2 biomolecules-13-00581-f002:**
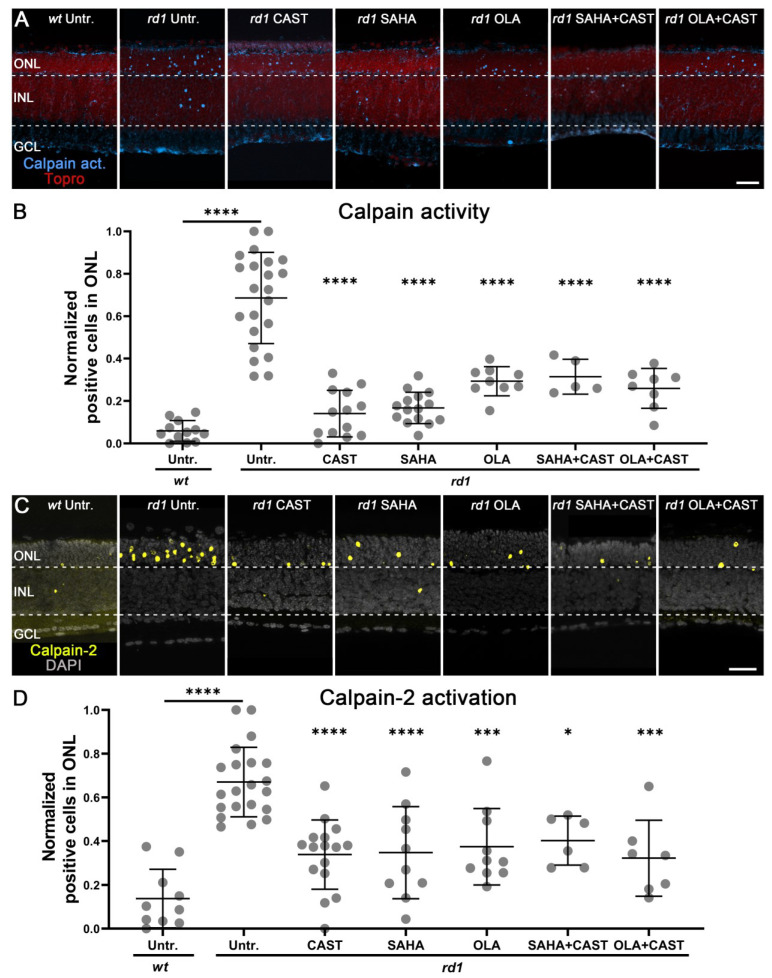
Effects of mono- and combination therapies on photoreceptor calpain activity. The activity of calpains and activation of the calpain-2 isoform were assessed in wild-type (*wt*) and *rd1* retinal explant cultures, treated with inhibitors targeting calpain (CAST), HDAC (SAHA), or PARP (OLA), or inhibitor combinations. (**A**) Retinal cross-sections from the different treatment groups were stained with the in situ calpain activity assay (blue). ToPro (red) was used for nuclear counterstaining. (**B**) Scatter plots showing the normalized percentages of calpain-positive cells in the outer nuclear layer (ONL) of treated or untreated (Untr.) retina. Note the strong increase in calpain activity in the *rd1* ONL when compared to *wt*. Both mono- and combination therapies significantly reduced *rd1* calpain activity. However, combination therapy did not produce additional benefits over monotherapy. (**C**) Immunostaining specific for activated calpain-2 (yellow). DAPI (grey) was used for nuclear counterstaining. (**D**) Quantification of activated calpain-2 positive cells in the ONL. Calpain-2 activation was markedly increased in the *rd1* ONL and significantly decreased by the various drug treatments. Images shown represent at least five different specimens for each genotype. For calpain activity: Untr. *wt*: 12 explants from 12 different mice; untr. *rd1*: 22/22; CAST *rd1*: 13/13; SAHA *rd1*: 15/15; OLA *rd1*:9/9; SAHA + CAST *rd1*:5/5; OLA + CAST *rd1*: 8/8; for calpain-2 activation: Untr. wt: 10 explants from 10 different mice; untr. *rd1*: 21/21; CAST *rd1*: 14/14; SAHA *rd1*: 10/10; OLA *rd1*: 10/10; SAHA + CAST *rd1*: 6/6; OLA + CAST *rd1*: 6/6; error bars represent SD; INL = inner nuclear layer; GCL = ganglion cell layer. Scale bar = 50 µm; significance levels: *, *p* < 0.05; ***, *p* < 0.001; ****, *p* < 0.0001.

**Figure 3 biomolecules-13-00581-f003:**
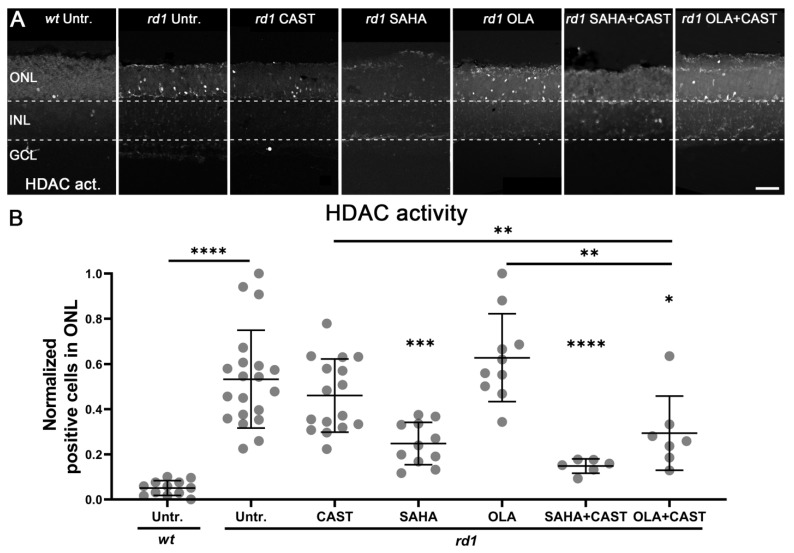
Effects of mono- and combination therapies on photoreceptor HDAC activity. (**A**) The retinal activity of HDACs was assessed using an in situ activity assay (grey). HDAC activity was strongly increased in the outer nuclear layer (ONL) of the untreated (Untr.) *rd1* retina compared to the wild-type (*wt*). (**B**) Quantification of the numbers of HDAC activity-positive cells in the various experimental groups. Unsurprisingly, the HDAC inhibitor SAHA significantly reduced HDAC activity; however, HDAC activity was not decreased by either the calpain inhibitor CAST or the PARP inhibitor OLA. The combination of OLA and CAST had a minor effect on HDAC activity. Images shown represent at least six different specimens for each genotype. Untr. *wt*: 12 explants from 12 different mice; untr. *rd1*: 20/20; CAST *rd1*: 16/16; SAHA *rd1*: 11/11; OLA *rd1*: 10/10; SAHA + CAST *rd1*: 6/6; OLA + CAST *rd1*: 7/7; error bars represent SD; INL = inner nuclear layer; GCL = ganglion cell layer. Scale bar = 50 µm; significance levels: *, *p* < 0.05; **, *p* < 0.01; ***, *p* < 0.001, ****; *p* < 0.0001.

**Figure 4 biomolecules-13-00581-f004:**
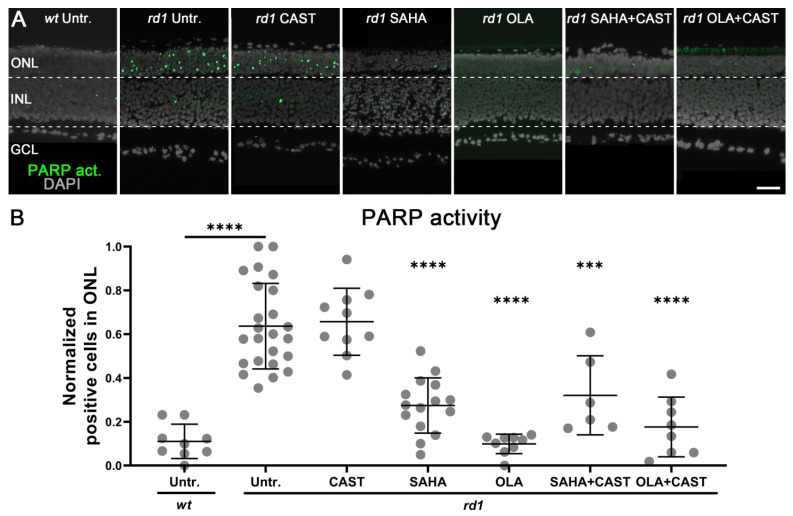
Effects of mono- and combination therapies on photoreceptor PARP activity. (**A**) Retinal PARP activity was assessed using an in situ activity assay (green). DAPI was used as nuclear counterstain (grey). In the outer nuclear layer (ONL) of the untreated (Untr.) *rd1* retina, PARP activity was strongly increased compared to the wild-type (*wt*). (**B**) Quantification of the numbers of PARP activity-positive cells in the various experimental groups. While the calpain inhibitor CAST had no effect on PARP activity, it was significantly reduced by both the HDAC inhibitor SAHA and the PARP inhibitor. Combined treatment with SAHA and CAST or OLA and CAST reduced PARP activity to levels comparable to monotherapy. Images shown represent at least six different specimens for each genotype. Untr. *wt*: 9 explants from 9 different mice; untr. *rd1*: 20/20; CAST *rd1*: 10/10; SAHA *rd1*: 15/15; OLA *rd1*: 9/9; SAHA + CAST *rd1*: 6/6; OLA + CAST *rd1*: 8/8; error bars represent SD; INL = inner nuclear layer; GCL = ganglion cell layer. Scale bar = 50 µm; significance levels: ***, *p* < 0.001; ****, *p* < 0.0001.

**Figure 5 biomolecules-13-00581-f005:**
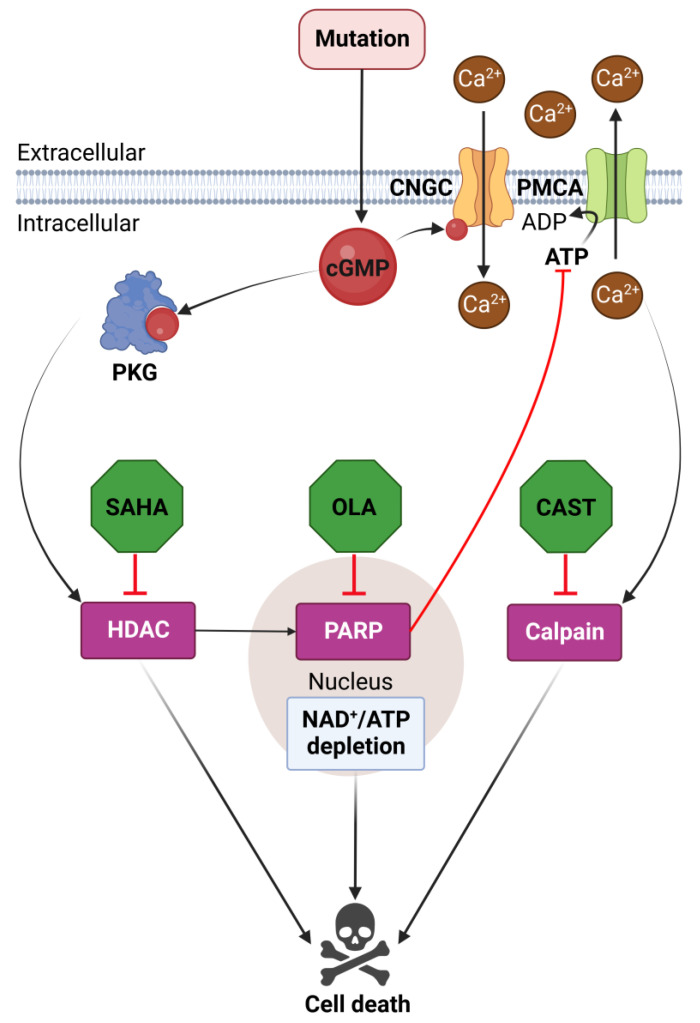
Molecular interactions between HDAC, PARP, and calpain during *rd1* retinal degeneration. In *rd1* photoreceptors, the initial genetic defect produces an accumulation of cGMP, which may then activate the cyclic nucleotide gated channel (CNGC) and protein kinase G (PKG). PKG, possibly in concert with voltage changes produced by CNGC, activates histone deacetylase (HDAC) resulting in DNA damage. Poly (ADP-ribose) polymerase (PARP) may attempt to repair such DNA damage, entailing a depletion of NAD^+^ and ATP. Since ATP is required for plasma membrane Ca^2+^ ATPase (PMCA)-mediated Ca^2+^ extrusion, loss of ATP leads to increased intracellular Ca^2+^ levels and excessive activation of calpain-type proteases. Eventually, the processes triggered by HDAC, PARP, and calpain activities result in photoreceptor cell death.

## Data Availability

Not applicable.

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
