# Peer review of "Inherited Retinal Degeneration: Towards the Development of a Combination Therapy Targeting Histone Deacetylase, Poly (ADP-Ribose) Polymerase, and Calpain"

_biomolecules, 2023, doi:10.3390/biom13040581_

Round 1
Reviewer 1 Report
In manuscript entitled “ Inherited retinal degeneration: Towards the development of a combination therapy targeting histone deacetylase, poly (ADP-ribose) polymerase, and calpain” the authors tried to explore the potential therapeutic effects of various combinations of HDAC, PARP, and calpain inhibitors on organotypic retinal explants obtained from wild-type mice and the rd1 mouse model for IRD. They employed various methods such as in situ activity assays, immunostaining, and the TUNEL assay to assess the efficacy of the treatments. The findings showed that inhibiting HDAC, PARP, or calpain significantly decreased photoreceptor degeneration in the rd1 mouse model, with the HDAC inhibitor Vorinostat (SAHA) being the most effective. The results also showed that combining either PARP and calpain inhibitors or HDAC and calpain inhibitors did not result in any additional photoreceptor protection. The manuscript holds potential for IRD treatment, however additional experimental evidence is necessary to support the hypothesis. Following are my comments/concerns/suggestions for the manuscript.
In the manuscript authors used single dose of different inhibitors as monotherapy and combine therapy I am just wondering if the authors performed a dose-dependent study to determine the selected concentrations of the inhibitors, such as 0.1 M SAHA, 1 M Olaparib, and 20 M Calpastatin, for monotherapy and same goes for combination therapy? Were the in-vitro cytotoxicity of each inhibitor evaluated?
What was the effect of single or combined treatment on wild-type mice group????
The authors should clearly state the number of mice utilized in each group for the study in the methodology section.
Authors used different combinational treatment groups for example SAHA and CAST or OLA and CAST, would not it be more interesting to add another group by combining SAHA, CAST and OLA and explore the effect???
Based on their findings authors demonstrated that HDAC controls both PARP and calpain activities, and that HDAC, PARP, and calpain likely are constituent members of the same degenerative pathway. Such conclusion cannot be made based on the experimental data presented in the manuscript. Further in vitro experimentations utilizing molecular biology techniques are required to validate this conclusion.
While the manuscript is written in understandable English, it is recommended that the authors have it reviewed by a native speaker in order to enhance the overall quality of the manuscript."
Reviewer 2 Report
This is a potentially very interesting study of the effects of inhibitors of calpains, PARP, and HDAC on a mouse model of inherited retinal degeneration (IRD). There are some issues with the approach which are potentially addressable, and whose resolution could potentially make the manuscript acceptable. The main issue is the selectivity of the inhibitors employed for the various isoforms of the targets. In particular, there are something like 18 PARP isoforms, 3 classes of HDACs, and multiple calpains, and it is not clear which are the most important in the various forms of IRD, for which 260-odd genes have been implicated. The HDACs and PARPs are metalloenzymes, with Zn in the active sites of HDACs, and the PARPs having Zn fingers, so it is not unreasonable that a hydroxamate inhibitor like SAHA that inhibited an HDAC by binding Zn might also inhibit a PARP; indeed, the authors found another alkyl hydroxamate, Trichostatin A, served as a PARP inhibitor in their Cell Death Disease 2010 paper. At a minimum I would suggest assessing the inhibition of SAHA on whatever PARP isozymes may be available as well as the HDACs, and similarly for OLA on the PARP and HDAC isozymes, and at least testing OLA and SAHA on the most available Calpain isozyme. It may be that (for instance) SAHA is a lousy inhibitor of the PARPs, and some HDACs, and this would strengthen the conclusions. It may also be that the narrow therapeutic window of SAHA mentioned by the authors reflects its (undesirable) inhibition of something else with a weaker Ki.
Other issues are that TUNEL doesn't measure cell death per se, but apoptosis at an advanced stage, and not necessarily other forms of cell death, particularly PARthanatos. Also, it would increase confidence in their method to accurately measure the numbers of cells by actual count in a few instances when assessing % positive, and compare with the number arrived at by measuring the average area: is it really 7 or 8 out of every thousand?
Reviewer 3 Report
In this paper the authors confirm that HDAC, PARP, and calpain may be therapeutic targets forinherited retinal degenerations treatment. They suggest that HDAC and PARP activity o, and CNGC and calpain activity form part of two branches of the same cell death pathway. Their initial hypothesis is that the inhibition of both branches should increase synergistically photoreceptor survival. To this end, they have used different combinations of inhibitors of these pathways. However the combination therapies tested did not produce the desired synergistic effect.
The topic is relevant and the manuscript is very well writte and clear. Figures are of high quality.
Despite the quality of the paper I suggest to make some modifications:
1. The authors claim in the conclusion that HDAC, PARP and calpain are integral to the same cell death pathway because they hace not obesrved synergistic interaction when combining treatments that target HDAC and calpain or PARP and calpain. The results do not support this conclusion. Though they suggest that this may be explained because the targets and mode of actions of the inhibitors are very differents, there are many other reasons that may explain this lack of synergistic effect. The authors should explain better their conclusion.
2. The authors don´t combine SAHA and Olaparib treatments. The affirm that PARP and HDACA are part of the same cell death branche. But it would be interesting to see the results with the combination of both treatments.
3. Review minor typo mistakes: for example line 349 and 350.
Round 2
Reviewer 1 Report
I have reviewed the revised version of the manuscript and I am pleased to report that the authors have significantly improved the quality of the paper. Furthermore, I am satisfied with the authors' responses to my previous comments and questions. Their revisions and clarifications have effectively addressed all of the issues I raised and have improved the overall clarity of the manuscript.